# Exact hydrodynamic solution of a double domain wall melting in the spin-1/2 $XXZ$ model

Stefano Scopa[1][*], Pasquale Calabrese[1,2] and Jérôme Dubail[3]

**1** SISSA and INFN, via Bonomea 265, 34136 Trieste, Italy
**2** International Centre for Theoretical Physics (ICTP), I-34151, Trieste, Italy
**3** Université de Lorraine, CNRS, LPCT, F-54000 Nancy, France

[*] sscopa@sissa.it

## Abstract

We investigate the non-equilibrium dynamics of a one-dimensional spin-1/2 XXZ model at zero-temperature in the regime $|\Delta| < 1$, initially prepared in a product state with two domain walls i.e, $|\downarrow \ldots \downarrow\uparrow \ldots \uparrow\downarrow \ldots \downarrow\rangle$. At early times, the two domain walls evolve independently and only after a calculable time a non-trivial interplay between the two emerges and results in the occurrence of a split Fermi sea. For $\Delta = 0$, we derive exact asymptotic results for the magnetization and the spin current by using a semi-classical Wigner function approach, and we exactly determine the spreading of entanglement entropy exploiting the recently developed tools of quantum fluctuating hydrodynamics. In the interacting case, we analytically solve the Generalized Hydrodynamics equation providing exact expressions for the conserved quantities. We display some numerical results for the entanglement entropy also in the interacting case and we propose a conjecture for its asymptotic value.

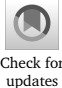

# 1 Introduction

A domain wall (DW) state of a quantum spin-1/2 chain is a product state prepared by joining two domains of aligned spins with opposite magnetization, as for instance $|\uparrow \dots \uparrow\downarrow \dots \downarrow\rangle$. Despite its simplicity, the time evolution of such a state shows non-trivial features of non-equilibrium dynamics and, for this reason, it has been the main character of a large number of studies, including stability analysis [1–3], exact computations for the free case e.g. [4–15], approximate and numerical results for the interacting integrable e.g. [16–29] and non-integrable e.g. [30–34] chain. For some integrable spin chains such as XXZ with anisotropy parameter $|\Delta| < 1$ (see Eq. (1) below), this research has led, only very recently, to an asymptotic analytical understanding of the DW dynamics e.g. [35–41], obtained by means of Generalized Hydrodynamics (GHD) [35, 42] and of its quantum fluctuations [43]. Though, some aspects of the melting process still lack of fully analytical explanations, such as the subleading corrections to the front dynamics [45] or the diffusive behavior at $|\Delta| \geq 1$ [46–49]. This exact solution substantiates an intuitive hydrodynamic picture at ballistic scales of the non-equilibrium dynamics which follows. Because of the integrability of the XXZ spin chain, stable quasiparticles excitations are initially emitted at the domain wall and propagate in time with a constant velocity, whose value depends on the interaction coupling. The fastest excitations of the spectrum define a fan-shaped spatio-temporal region (called *lightcone*) around the junction, inside which entanglement spreads and the initially ordered domains melt. Quite interestingly, the physics of the melting process is not modified by the presence of interactions. In particular, it has been shown that, up to a rescaling of the lightcone, the charges profiles in the gapless XXZ model [40] have the same behavior as a free Fermi system [4]. This similarity extends even to the half-system entanglement entropy, which shows the same asymptotic growth as $S_1(0,t) \sim \frac{1}{6} \log t$ in the free [7] and interacting [41] case. Notice that this is a peculiarity of DW states, related to the structure of its Bethe Ansatz equations. Indeed, in generic integrable models, interactions are responsible for a dressing of the observables that typically modifies the non-equilibrium dynamics, see e.g. Ref. [44, 50–52] for recent reviews. The case of bipartite spin states with initial correlations has also been considered in literature [6, 20, 21, 53], where it is observed that the presence of initial entangled quasiparticles results into a faster spreading of the half-system entanglement.

    In this work, we extend the rich analysis on DW states to the case where three domains of aligned spins with different orientations are joined together, i.e., for an initial state like $|\downarrow \dots \downarrow\uparrow \dots \uparrow\downarrow \dots \downarrow\rangle$. In the following, we will refer to this state as *double domain wall*. Intuitively, the non-equilibrium dynamics of a XXZ spin chain prepared in a double DW state can be split into two regimes. At early times, the light cones generated at the two junctions do not intersect and the evolution is that of two independent DW states of Ref. [40]. Conversely, when the quasiparticles emitted from the two walls meet, a non-trivial interplay between the physics of the single DWs takes place and characterizes the melting process at later times. In this regime, the half-system entanglement (i.e., with the entangling point in the middle of the central domain, which is initially equal to zero) is fed by the double contribution of

left and right moving quasiparticles coming from the two walls and therefore it undergoes a rapid growth. At large times, the entanglement saturates to a constant value $S_1(0, t) \sim \frac{1}{3} \log \ell$, which depends on the size $2\ell$ of the domain of upwards spins at $t = 0$. Throughout the rest of the work, we present exact calculations and numerical analysis in order to corroborate these intuitions with quantitative arguments.

*Outline.* We organize the contents of the paper as follows. In Sec. 2, we introduce the model and we set up the quench protocol considered in this work. Although the physics of the melting process is found to be qualitatively similar for any value of the interaction coupling $|\Delta| < 1$, we treat the non-interacting case ($\Delta = 0$) in a dedicated section (Sec. 3), for a clearer exposition. In particular, in Sec. 3.1, we characterize the semi-classical evolution in phase space in terms of the Wigner function and we derive exact asymptotic results for the magnetization and the spin current profiles. In Sec. 3.2 we discuss the behavior of the entanglement. As pointed out in the recent works [6, 41, 43, 54], this study requires a re-quantization of the semi-classical hydrodynamic background in terms of a Luttinger liquid and the use of conformal invariance. Exact numerical lattice calculations are performed to test and complement our findings for the free model. Section 4 contains instead the analysis of the model at finite interactions. After a short summary of the Bethe Ansatz solution in Sec. 4.1, in Sec. 4.2 we consider the hydrodynamic limit of the spin chain and we characterize the initial state at $t = 0$ using the local density approximation. The GHD equations are reported in Sec. 4.3, together with a detailed derivation of their analytic solution. The exact computation of the profiles follows in Sec. 4.4, where one can also find numerical checks based on time-dependent Density Matrix Renormalization Group (tDMRG). The analytic treatment of the entanglement evolution in the interacting case is very challenging as it needs a careful analysis based on quantum GHD [43]. This study goes well beyond the goal of this paper. Nevertheless, in Sec. 4.5, we present a numerical analysis of the entanglement and we compare our numerical findings with the exact solution of the non-interacting case. Finally, we report our conclusions in Sec. 5 and some technical aspects of the Bethe Ansatz solution in Appendix A.

## 2 Setup of the problem

We consider the one-dimensional XXZ model with Hamiltonian

$$\hat{H} = -\frac{1}{4} \sum_{x=-L/2}^{L/2-1} \left( \hat{\sigma}_x^x \hat{\sigma}_{x+1}^x + \hat{\sigma}_x^y \hat{\sigma}_{x+1}^y + \Delta \hat{\sigma}_x^z \hat{\sigma}_{x+1}^z \right), \tag{1}$$

where $L$ is the length of the chain and $\hat{\sigma}_x^{\alpha=x,y,z}$ are standard Pauli operators acting on site $x$. Here, $\Delta$ is the interaction coupling which we set to be in the gapless regime $|\Delta| \leq 1$, where it is customary to write $\Delta = \cos\gamma$. We do not consider $|\Delta| > 1$ because energy arguments show that domain wall states do not melt, see Ref. [46, 47]. The case $|\Delta| = 1$ is very peculiar [48, 49] and therefore it will be also excluded from this study. We further focus on the rational case, i.e., on those values of $\gamma$ that can be written as the ratio $\gamma = \pi Q/P$, with $Q$, $P$ two co-prime integers, $1 \leq Q < P$. In the rational case, $\gamma$ admits a continued fraction representation

$$\gamma = \cfrac{\pi}{\nu_1 + \cfrac{1}{\nu_2 + \cfrac{1}{\nu_3 + \ldots}}}, \tag{2}$$

where $\{\nu_1, \ldots, \nu_q\}$ is a set of numbers satisfying $\nu_1, \ldots, \nu_{q-1} \geq 1$ and $\nu_q \geq 2$. In the thermodynamic limit $L \to \infty$, the model (1) is exactly solved by means of Thermodynamic Bethe Ansatz (TBA). In particular, for large values of the system size $L$ and under the string hypothesis [55], one can describe the excitation spectrum of the spin chain in terms of different species of

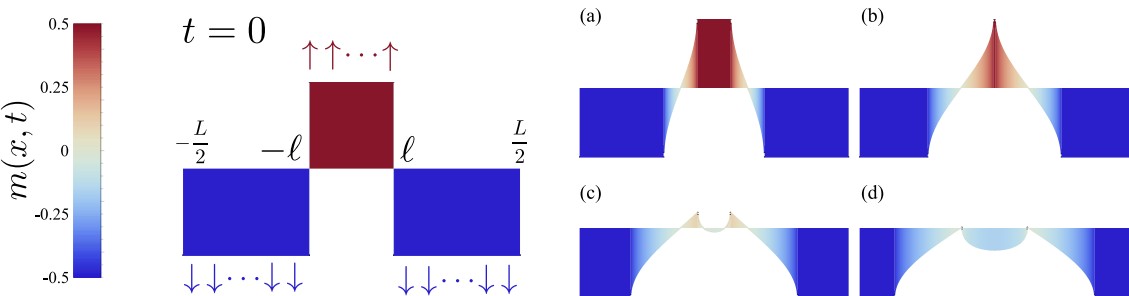

Figure 1: Illustration of the melting dynamics of the double domain wall state in Eq. (4). At $t = 0$ the system is prepared in the product state of three ordered domains, resulting into a step-like shape of the magnetization profile $m(x, t)$ (left panel). (a) At early times, the central domain of aligned spin begins to melt into the left and right ferromagnets generating non-homogeneous propagation fronts around the junctions. After the initial transient regime, the two propagating fronts merge (b) and contribute together to the relaxation (c)-(d).

quasiparticles (generically referred to as *strings*). A short summary of the TBA solution of the model (1) will be reported in Sec. 4.1 while we address the reader to e.g. Ref. [55, 56] for a comprehensive discussion. The total number of allowed strings $\delta$ is read from the interaction coupling in Eq. (2) as

$$\delta = \sum_{k=1}^{q} \nu_k \,. \tag{3}$$

Each of the strings has a well-defined *length* $l_j$, *parity* $u_j$ and *sign* $s_j$, which are all given in terms of $\{\nu_1, \ldots, \nu_q\}$ after some manipulations [40, 55], see Appendix A.

At $t = 0$, we prepare the system in the product state

$$|\Psi(0)\rangle = \bigotimes_{x=-L/2}^{-\ell-1} |\downarrow_x\rangle \otimes \bigotimes_{x=-\ell}^{\ell} |\uparrow_x\rangle \otimes \bigotimes_{x=\ell+1}^{L/2} |\downarrow_x\rangle \,, \tag{4}$$

where $\pm \ell$ are the positions of the domain walls and $|\uparrow_x\rangle$ (resp. $|\downarrow_x\rangle$) is the eigenstate of $\hat{\sigma}_x^z$ with eigenvalue $+1$ (resp. $-1$). For $t > 0$ we consider the Hamiltonian dynamics generated by (1)

$$|\Psi(t)\rangle = e^{-it\hat{H}} |\Psi(0)\rangle \,, \tag{5}$$

during which the central ordered domain of (4) gradually melts into the left and right ferromagnets, see Fig. 1 for an illustration.

Our focus will be on the hydrodynamic limit of the model, where several exact results can be derived in the limit of large space and time scales. Specifically, we shall consider the scaling limit $x, \ell, L, t \to \infty$ at fixed ratios $x/t$ and $\ell/L$ and we shall investigate the non-equilibrium dynamics of the local magnetization $m$ and of the local spin current $J$, defined as

$$m(x, t) = \frac{1}{2} \langle \Psi(t) | \hat{\sigma}_x^z | \Psi(t) \rangle \,, \qquad J(x, t) = \frac{1}{4} \langle \Psi(t) | \hat{\sigma}_x^y \hat{\sigma}_{x+1}^x - \hat{\sigma}_x^x \hat{\sigma}_{x+1}^y | \Psi(t) \rangle \,. \tag{6}$$

The initial state (4) has zero entanglement for any bipartition. However, the latter is generated during the melting dynamics through the spreading of quasiparticles from the junctions towards the ordered regions. Therefore, we will investigate the formation and the subsequent growth of the entanglement entropy by focusing on the Rényi entropy of a subsystem $A = [x, +\infty]$

$$S_\alpha(x, t) = \frac{1}{1-\alpha} \log \operatorname{tr} (\hat{\rho}_A)^\alpha \tag{7}$$

and its limit $\alpha \to 1$, i.e., the Von Neumann entanglement entropy

$$S_1(x,t) = -\operatorname{tr}\hat{\rho}_A(t)\log\hat{\rho}_A(t). \tag{8}$$

In Eqs. (7), (8), $\hat{\rho}_A = \operatorname{tr}_{\bar{A}}\hat{\rho}(t)$ is the reduced density matrix of the subsystem $A$, obtained by tracing out the degrees of freedom of the interval $\bar{A} = [-\infty, x)$ from the full density matrix $\hat{\rho} = |\Psi(t)\rangle\langle\Psi(t)|$.

## 3 Analytic solution of the non-interacting case

We first investigate the melting dynamics of the double DW (4) in a non-interacting spin chain, leaving the discussion of the interacting model to Sec. 4. After setting $\Delta = 0$ in Eq. (1), the spin chain Hamiltonian reduces, up to an irrelevant additive constant, to the free Fermi gas

$$\hat{H} = -\frac{1}{2}\sum_{x=-L/2}^{L/2-1}\left(\hat{c}_x^\dagger\hat{c}_{x+1} + \hat{c}_{x+1}^\dagger\hat{c}_x\right), \tag{9}$$

where $\hat{c}_x^\dagger$, $\hat{c}_x$ are standard lattice Fermi operators, obtained from the ladder Pauli operators $\hat{\sigma}_x^\pm = (\hat{\sigma}_x^x \pm \mathbf{i}\hat{\sigma}_x^y)/2$ after a Jordan-Wigner transformation [57]. Although the non-equilibrium evolution of the free Fermi gas (9) can be investigated by means of exact lattice calculations [4,5], we shall focus on a hydrodynamic limit. Indeed, a large-scale description gives not only access to asymptotically exact results for several quantities of interest (such as the magnetization profile and the spin current) without difficult computations, but also, it allows us to characterize the entanglement evolution during the melting process, whose lattice derivation is currently out-of-reach with standard techniques.

### 3.1 Semi-classical hydrodynamics

In the thermodynamic limit $L, \ell \to \infty$ at fixed ratio $\ell/L$, we describe the local physics of the model (9) in terms of coarse-grained cells, each containing a large number of lattice sites. Inside each cell, we diagonalize the Hamiltonian (9) in Fourier space, obtaining [6]

$$\hat{H} = -\int_{-L/2}^{L/2}\mathrm{d}x\int_{-\pi}^{\pi}\frac{\mathrm{d}k}{2\pi}\,\cos k\,\hat{\eta}_{k,x}^\dagger\,\hat{\eta}_{k,x}, \tag{10}$$

where the lattice index $x$ is replaced by a continuous variable. Here, $\hat{\eta}_{k,x}^\dagger$, $\hat{\eta}_{k,x}$ are the creation and annihilation operators of a fermionic particle with momentum $k$ inside the cell labeled by $x$. In this limit, it is easy to see that the initial state in Eq. (4) corresponds to a gas where fermionic particles with momentum $-\pi \leq k \leq \pi$ entirely fill the spatial region $-\ell \leq x \leq \ell$, leaving empty the rest of the chain. Therefore, in terms of the Wigner function $\mathrm{n}(x,k)$ [58,59] (which is essentially the occupation function of the free Fermi gas), the macrostate at $t = 0$ corresponding to the double DW state (4) is given by

$$\mathrm{n}(x,k) = \begin{cases} 1, & \text{if } |x| \leq \ell \text{ and } |k| \leq \pi, \\ 0, & \text{otherwise}. \end{cases} \tag{11}$$

At times $t > 0$, the fermionic particles propagate independently with constant velocity $v(k) = \sin k$. This implies that the occupation number at time $t > 0$ can be obtained by tracking backward the trajectory of each particle up to $t = 0$ and it reads

$$\mathrm{n}(x,t,k) = \mathrm{n}(x - t\sin k, 0, k). \tag{12}$$

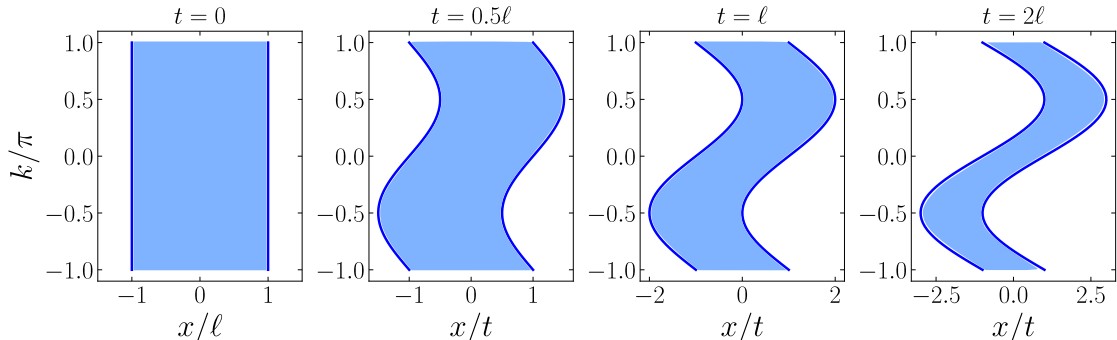

Figure 2: Snapshots of the Wigner function (12) during the melting dynamics of the double DW (11).

The result (12) might be also viewed as solution of a Euler-like hydrodynamic equation [6, 54]

$$(\partial_t + \sin k \; \partial_x) n(x, t, k) = 0, \tag{13}$$

satisfied by the Wigner function, at lowest order in the $\partial_x$ and $\partial_k$ derivatives [60–63]. A microscopic derivation of Eq. (13) for non-interacting fermions can be found in Refs. [60, 61]. Similar studies in higher spatial dimensions $d > 1$ and at non-zero temperature can be found in Ref. [64, 65]. In Fig. 2, we show the hydrodynamic evolution of $n(x, t, k)$ during the double DW melting process.

Notice that the Wigner function in Eq. (12) shows a particularly simple form as it takes only the values 0 or 1, due to the zero-entropy condition of the local states during the time evolution [66]. As a consequence, it is possible to encode all the information about the phase-space dynamics in terms of the contour of the Wigner function, typically called *Fermi contour* $\Gamma(t)$, which keeps track of the Fermi points at each position $x$.

From the inspection of Fig. 2, one can graphically determine the Fermi points at each $x$ and $t$. In particular, the number $n$ of Fermi points is given by the number of intersections of a vertical line drawn at position $x$ and time $t$ with the contour of $n(x, t, k)$. With this method, we observe that two Fermi points are found at each position $x$ as long as $t < \ell$ while, for $t \geq \ell$, one enters in a richer landscape where split Fermi seas can be found. This behavior of the Fermi seas can be easily understood considering lightcone regions $||x| - \ell|/t \leq 1$ around the two junctions $x = \pm \ell$, determined by the propagation of the fastest modes with $k = \pm \pi/2$ and velocity $v(\pm \pi/2) = \pm 1$, see Fig. 3(a) for an illustration. At times $t < \ell$, the dynamics of the particles inside the two lightcones is independent and thus we find a connected Fermi sea at any position $x$. In this regime, the evolution is that of two independent domain walls [4]. Conversely, at times $t \geq \ell$, the two lightcones intersect over the spatial region $x \in [-t + \ell, t - \ell]$ and, as a consequence, a split Fermi sea is found. Precisely, the number $n$ of Fermi points in each regime is

$$n = \begin{cases} 4, & \text{if } |x| \leq t - \ell; \\ 2, & \text{if } |t - \ell| < |x| \leq t + \ell; \\ 0, & \text{otherwise.} \end{cases} \tag{14}$$

Since the propagating modes on the Fermi contour are only those initially located at $x = \pm \ell$, the analytic expression of the Fermi points follows from the equation of motion

$$x = \pm \ell + \sin k_F \; t, \tag{15}$$

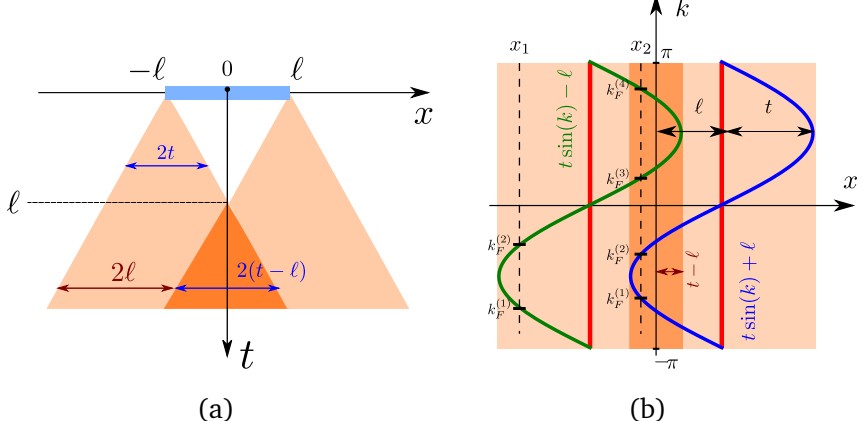

(a)  (b)

Figure 3: (a) Illustration of the particles spreading during the melting dynamics. From the junctions at $x = \pm\ell$, two lightcones open up and determine the melted region $||x| - \ell| < t$. At early times $t < \ell$ these lightcones do not intersect and a connected Fermi sea is found at any position $x$ (light-orange region). Conversely, when $t \geq \ell$, the particles emitted from one junction penetrate inside the lightcone of the other, leading to a split Fermi sea for $|x| \leq t - \ell$ (dark-orange region). (b) Graphical solution of Eq. (15) at fixed time $t$: for any $x_1$ such that $|t - \ell| < |x_1| \leq t + \ell$ (light-orange region), one obtains two Fermi points $k_F^{(1,2)}(x_1, t)$, while for $|x_2| \leq t - \ell$ (dark-orange region) one finds the four solutions $k_F^{(s)}(x_2, t)$, $s = 1, \ldots, 4$. In each regime, the values of the roots $k_F^{(s)}(x, t)$ are given in Eq. (17). Thick red lines denotes the intial position $\pm\ell$ of the two domain walls.

which is readily solved as

$$k_F = \left\{ \arcsin \frac{x \pm \ell}{t}; \pi - \arcsin \frac{x \pm \ell}{t} \right\}. \tag{16}$$

By exploiting the symmetry properties of the contour and with elementary algebra, one can show that the four roots in Eq. (16) combine together to give the following Fermi points in each regime:

$$k_F(x, t) = \begin{cases} \pm \arcsin \frac{||x| - \ell|}{t}; \pm \left( \pi - \arcsin \frac{||x| - \ell|}{t} \right), \\ \quad \text{if } |t - \ell| < |x| \leq t + \ell; \\ \pm \arcsin \frac{|x| + \ell}{t}; \pm \left( \pi - \arcsin \frac{|x| + \ell}{t} \right); \mp \arcsin \frac{||x| - \ell|}{t}; \mp \left( \pi - \arcsin \frac{||x| - \ell|}{t} \right), \\ \quad \text{if } |x| \leq t - \ell, \end{cases} \tag{17}$$

with sign determined by

$$\text{sign}(k_F(x, t)) = \text{sign}(x(|x| - \ell)), \tag{18}$$

as illustrated in Fig. 3(b). Denoting such Fermi points with $k_F^{(s)}(x, t)$ ($s = 1, \ldots, n$ and $k_F^{(s)}$ in increasing order), and the (split-)Fermi sea $\Gamma(x, t)$ with

$$\Gamma(x, t) = \bigcup_{s=1}^{n-1} [k_F^{(s)}(x, t), k_F^{(s+1)}(x, t)], \qquad \left( \Gamma(t) = \bigcup_x \Gamma(x, t) \right) \tag{19}$$

one can derive exact asymptotic results by summing up each individual contribution coming from a filled mode at position $x$ and time $t$, i.e., for each $k \in \Gamma(x, t)$. For instance, the

asymptotic result for the magnetization profile is given by

$$m(x,t) \;\; = -\tfrac{1}{2} + \int_{\Gamma(x,t)} \tfrac{dk}{2\pi} = -\tfrac{1}{2} + \sum_{s=1}^{n-1} \tfrac{k_F^{(s+1)}(x,t) - k_F^{(s)}(x,t)}{2\pi} \,. \tag{20}$$

Explicitly, using Eq. (17) in (20), we obtain

$$m(x,t) = \begin{cases} -\tfrac{1}{2} + \tfrac{1}{\pi}\left(\arcsin\tfrac{|x|+\ell}{t} - \arcsin\tfrac{|x|-\ell}{t}\right), & \text{if } |x| \le t - \ell\,; \\ -\tfrac{1}{\pi}\arcsin\tfrac{|x|-\ell}{t}, & \text{if } |t-\ell| < |x| \le t+\ell\,. \end{cases} \tag{21}$$

Similarly, the spin current in Eq. (6) is obtained as the weighted sum

$$J(x,t) = \int_{\Gamma(x,t)} \tfrac{dk}{2\pi} \sin k = \tfrac{1}{2\pi} \sum_{s=1}^{n-1}\left(\cos k_F^{(s)}(x,t) - \cos k_F^{(s+1)}(x,t)\right) \tag{22}$$

$$= \begin{cases} \tfrac{\text{sign}(x)}{\pi}\left(\sqrt{1 - \tfrac{(|x|-\ell)^2}{t^2}} - \sqrt{1 - \tfrac{(|x|+\ell)^2}{t^2}}\right), & \text{if } |x| \le t-\ell\,; \\ \tfrac{\text{sign}(x)}{\pi}\sqrt{1 - \tfrac{(|x|-\ell)^2}{t^2}}, & \text{if } |t-\ell| < |x| \le t+\ell\,. \end{cases} \tag{23}$$

In Fig. 4, we show the exact asymptotic formulae for the profiles in Eq. (21)-(22) against exact lattice numerical calculations finding a perfect agreement. The numerical data are obtained from the lattice evolution of the two-point function

$$G_{x,x'}(t) = \langle \Psi(t) | \hat{c}_x^\dagger \hat{c}_{x'} | \Psi(t) \rangle \,. \tag{24}$$

In particular, we determine $G_{x,x'}(0)$ from the exact diagonalization of the Hamiltonian

$$\hat{H}_{xx}^{(0)} = -\frac{1}{2}\sum_{x=-L/2}^{L/2-1}\left(\hat{c}_x^\dagger \hat{c}_{x+1} + \text{h.c.}\right) + \sum_{x=-L/2}^{L/2} V(x)\hat{c}_x^\dagger \hat{c}_x \tag{25}$$

with auxiliary potential

$$V(x) = \begin{cases} -\Lambda, & \text{if } |x| \le \ell\,; \\ \Lambda, & \text{otherwise}\,, \end{cases} \qquad \Lambda \to \infty\,, \tag{26}$$

which is chosen such that its ground state reproduces the initial state of Eq. (4). Subsequently, we evolve the two-point function in the eigenstate basis of the Hamiltonian (9), see Ref. [6,20] for a detailed discussion on the numerical implementation. From the exact evolution of the two-point correlation (24), it follows that

$$m(x,t) = -\frac{1}{2} + G_{x,x}(t), \qquad J(x,t) = \frac{1}{2i}\left(G_{x,x+1}(t) - G_{x+1,x}(t)\right). \tag{27}$$

## 3.2 Quantum fluctuations and Entanglement entropy

It is important to notice that the phase-space approach discussed in Sec. 3.1 does not include quantum fluctuations of the Fermi contour and therefore, it is not sufficient to characterize the entanglement spreading during the melting dynamics. On the other hand, an exact lattice calculation for such non-homogeneous and non-equilibrium processes is currently not possible with standard techniques, even for the free model (9) considered in this section. To overcome this limit, it has been recently noticed [6, 41, 43, 54] that the asymptotic behavior of the entanglement can effectively described by a quantum hydrodynamic theory, obtained after

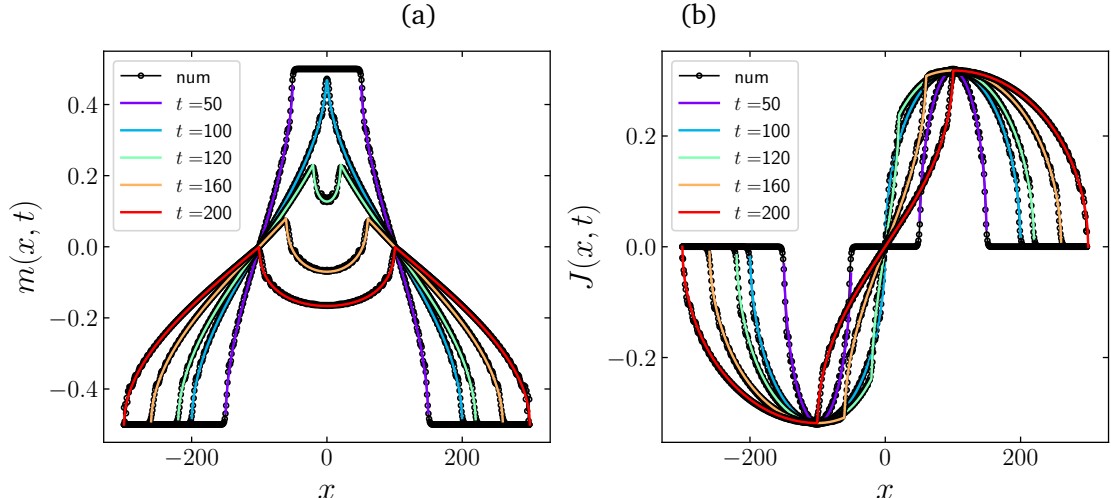

Figure 4: (a) Magnetization and (b) spin current profiles for the non-interacting double domain wall melting problem. The different curves show the behavior of the analytic solutions (21)-(22) at different times while the symbols are obtained with exact numerical calculations on a lattice of size $L = 600$ and $\ell = 100$. The matching of the profiles with the numerics is extremely good.

the re-quantization of the Fermi contour of Sec. 3.1 in terms of a Luttinger-liquid [67]. The latter has to reproduce the relevant low-energy quantum physics in the vicinity of the Fermi points, which essentially consists in the formation of particle-hole excitations around the Fermi contour. Following this program, we consider the quantum fluctuations of the fermionic density in terms of a fluctuating-field $\hat{\chi}$

$$\delta\hat{\varrho}(x,t) = \frac{1}{2\pi}\partial_x\hat{\chi}(x,t) \tag{28}$$

and we expand, at leading order in the scaling dimension of the low-energy fields, the time-dependent creation and annihilation fermionic operator with standard bosonization method [67–69]

$$\hat{c}_x^\dagger(t) \propto\; :\exp\left(\frac{\mathrm{i}}{2}\left[\hat{\chi}_+(x,t)-\hat{\chi}_-(x,t)\right]\right):\, + \text{less relevant operators}\,;$$
$$\hat{c}_x(t) \propto\; :\exp\left(\frac{\mathrm{i}}{2}\left[\hat{\chi}_-(x,t)-\hat{\chi}_+(x,t)\right]\right):\, + \text{less relevant operators}\,, \tag{29}$$

up to a non-universal amplitude and a semi-classical phase which are not important for our scopes (see e.g. Ref. [6, 7, 54] for more details on the phase and e.g. Ref. [70, 71] for the amplitude). In Eq. (29), we have introduced the chiral components of the fluctuating field $\hat{\chi} = \hat{\chi}_+ + \hat{\chi}_-$ which carry the physical interpretations of right (+) and left (−) moving excitations along the Fermi contour. It is therefore useful to introduce a parametrization of the Fermi contour $\Gamma(t)$ in terms of a coordinate $\theta$ along the curve as

$$\Gamma(t) = \Big\{(x(\theta),k(\theta)):\; k(\theta)\in\Gamma(x(\theta),t)\Big\}. \tag{30}$$

The dynamics of the chiral fields $\hat{\chi}_\pm$ is then governed by the conformal field theory of a free massless boson or, equivalently, by the Luttinger-liquid Hamiltonian along the Fermi contour (see e.g. [6, 41, 43, 54, 71, 72])

$$\hat{H}_{LL}[\Gamma] = \int_\Gamma \frac{\mathrm{d}\theta}{2\pi}\, \mathcal{J}(\theta)\sin k(\theta)\, (\partial_\theta\hat{\chi}_a)^2\,, \tag{31}$$

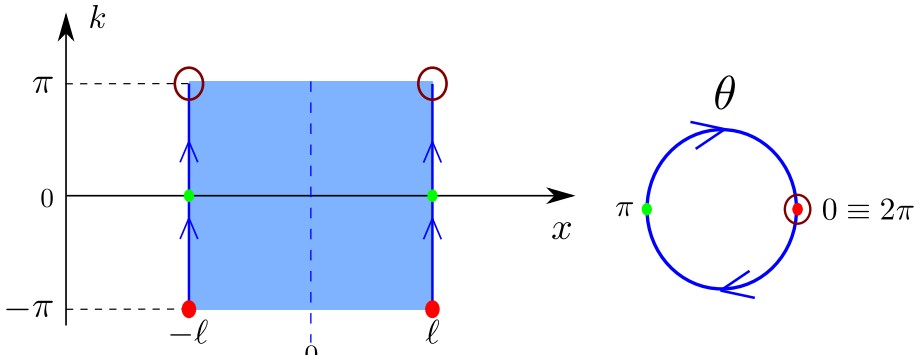

Figure 5: Parametrization of the initial Fermi contour with the coordinate $\theta$ along the unit circle. Notice that the horizontal branches of the contour do not contain propagating modes and therefore are not needed for the description of the entanglement evolution.

where $\mathcal{J}(\theta)$ is a jacobian factor and $a = \pm$ if $k(\theta) \gtrless 0$. Importantly, in our hydrodynamic description of the problem, quantum fluctuations in the initial state are given by the ground state of $\hat{H}_{LL}[\Gamma(0)]$ and the effect of the time-evolution is simply to transport such quantum fluctuations along the curve $\Gamma(t)$, which gets deformed over time according to the semi-classical dynamics of Sec. 3.1.

With this framework, we now consider the large-scale contribution to the Rényi entropy in Eq. (7) of a subsystem $A = [x, +\infty]$ by using a conformal field theory (CFT) approach. For integer Rényi index $\alpha$, the latter can be expressed as the expectation value of a twist field $\hat{\mathcal{T}}_\alpha$ living at the boundaries of the subsystem $A$ [73–75]

$$\tilde{S}_\alpha(x, t) = \frac{1}{1-\alpha} \log \langle \mathcal{T}_\alpha(x, t) \rangle . \tag{32}$$

Moreover, in our chiral model, $\hat{\mathcal{T}}_\alpha$ admits a decomposition in chiral twist fields $\{\hat{\Phi}_\alpha^-, \hat{\Phi}_\alpha^+\}$ that behave under conformal mappings as primary fields with scaling dimension

$$h_\alpha = \frac{1}{24}\left(\alpha - \frac{1}{\alpha}\right). \tag{33}$$

It follows that $\tilde{S}_\alpha(x, t)$ can be written as the $n$-point correlation function of the chiral fields $\hat{\Phi}_\alpha^-, \hat{\Phi}_\alpha^+$ of the CFT which lives along the Fermi contour at time $t$. Because of the conservation of momentum during the time-evolution, the Fermi points $k_F^{(s)}(x, t)$ at time $t$ can be traced backward to the initial Fermi contour where they can be simply parametrized as

$$\theta = \pi + k \quad (-\pi \le k \le \pi), \tag{34}$$

as illustrated in Fig. 5. Denoting with $\theta_s$ the positions of $k_F^{(s)}(x, t)$ along the initial Fermi contour, the Rényi entropy reduces to

$$\tilde{S}_\alpha(x, t) = \frac{1}{1-\alpha} \log \left[ \prod_{s=1}^{n} \left| \frac{d\theta}{dx} \right|_{\theta=\theta_s}^{h_\alpha} \left\langle \prod_{q=0}^{n/2-1} \hat{\Phi}_\alpha^+(\theta_{2q+1})\hat{\Phi}_\alpha^-(\theta_{2q+2}) \right\rangle \right] \tag{35}$$

and it can be exactly determined, as we now discuss. From Eq. (16) we obtain the Weyl factors

$$\prod_{s=1}^{n} \left| \frac{d\theta}{dx} \right|_{\theta=\theta_s} = \begin{cases} \left[ t\sqrt{1 - \frac{(|x|-\ell)^2}{t^2}} \right]^{-2}, & \text{if } |t-\ell| < |x| \le t+\ell ; \\[4mm] \left[ t^2 \sqrt{1 - \frac{(|x|-\ell)^2}{t^2}} \sqrt{1 - \frac{(|x|+\ell)^2}{t^2}} \right]^{-2}, & \text{if } |x| \le t-\ell , \end{cases} \tag{36}$$

while the $n$-point correlation function is given by

$$\left\langle \prod_{q=0}^{n/2-1} \hat{\Phi}_\alpha^+(\theta_{2q+1}) \hat{\Phi}_\alpha^-(\theta_{2q+2}) \right\rangle = \begin{cases} g(\theta_1, \theta_2), & \text{if } |t-\ell| < |x| \le t+\ell; \\ \frac{g(\theta_1,\theta_2)g(\theta_3,\theta_4)g(\theta_1,\theta_4)g(\theta_2,\theta_3)}{g(\theta_1,\theta_3)g(\theta_2,\theta_4)}, & \text{if } |x| \le t-\ell, \end{cases} \tag{37}$$

where

$$g(\theta, \theta') = \left| 2\sin\frac{\theta-\theta'}{2} \right|^{-2h_\alpha}. \tag{38}$$

Plugging Eq. (36) and (37) into Eq. (35) and with simple algebra, we arrive to the result

$$\tilde{S}_\alpha(x,t) = \begin{cases} \frac{\alpha+1}{12\alpha} \log\left( 2t^2 \cos^2\phi \, \cos^2\varphi \, \frac{\sin^2\left(\frac{\phi-\varphi}{2}\right)}{\cos^2\left(\frac{\phi+\varphi}{2}\right)} \right), & \text{if } |x| \le t-\ell; \\ \frac{\alpha+1}{12\alpha} \log(4t\cos^2\phi), & \text{if } |t-\ell| < |x| \le t+\ell, \end{cases} \tag{39}$$

where we introduced the shorthands

$$\phi \equiv \arcsin\frac{|x|-\ell}{t}; \qquad \varphi \equiv \arcsin\frac{|x|+\ell}{t}. \tag{40}$$

The CFT result in Eq. (39) provides only the universal contribution to the Rényi entropies and it has to be complemented with a non-universal cutoff $\epsilon(x,t)$ so that

$$S_\alpha(x,t) = \frac{1}{1-\alpha} \log\left[ \epsilon^{2h_\alpha} \langle \hat{\mathcal{T}}(x,t) \rangle \right] = \tilde{S}_\alpha(x,t) + \frac{2h_\alpha}{1-\alpha} \log\epsilon(x,t). \tag{41}$$

For a non-interacting spin-1/2 chain, the exact expression of the cutoff for the von-Neumann entanglement entropy is known [73, 76]. In particular, for $n = 2$, i.e., for a connected Fermi sea, it is given in terms of the magnetization profile in Eq. (20) as [6, 7]

$$\epsilon(x,t) = \frac{C}{\cos\pi m(t,x)} = C/\cos\phi, \qquad \text{if } |t-\ell| < |x| \le t+\ell, \tag{42}$$

where $C$ is a known non-universal amplitude, see Ref. [76] and Eq. (46) below. In the presence of the split Fermi sea, the computation of $\epsilon$ is more involved but it can be still performed analytically exploiting the Fisher-Hartwig conjecture. As a result, one obtains the generic formula [77]

$$\log\epsilon(x,t) = \sum_{1 \le s < s' \le n} (-1)^{s+s'} \log\left| \sin\frac{\theta_s - \theta_{s'}}{2} \right| + \frac{n}{2} \log C. \tag{43}$$

One can check that Eq. (43) correctly reproduces the cutoff in Eq. (42) for $n = 2$ while, in the regime with four Fermi points, it gives

$$\epsilon(x,t) = \frac{C^2}{\cos\phi\cos\varphi}, \qquad \text{if } |x| \le t-\ell. \tag{44}$$

Hence, including the cutoff contribution (42)-(44) in Eq. (39) and taking the replica limit $\alpha \to 1$, we finally obtain the result for the entanglement entropy

$$S_1(x,t) = \begin{cases} \frac{1}{6} \log\left( t^2 |\cos^3\phi| \, |\cos^3\varphi| \, \frac{\sin^2\left(\frac{\phi-\varphi}{2}\right)}{\cos^2\left(\frac{\phi+\varphi}{2}\right)} \right) + 2\Upsilon, & \text{if } |x| \le t-\ell; \\ \frac{1}{6} \log(t|\cos^3\phi|) + \Upsilon, & \text{if } |t-\ell| < |x| \le t+\ell, \end{cases} \tag{45}$$

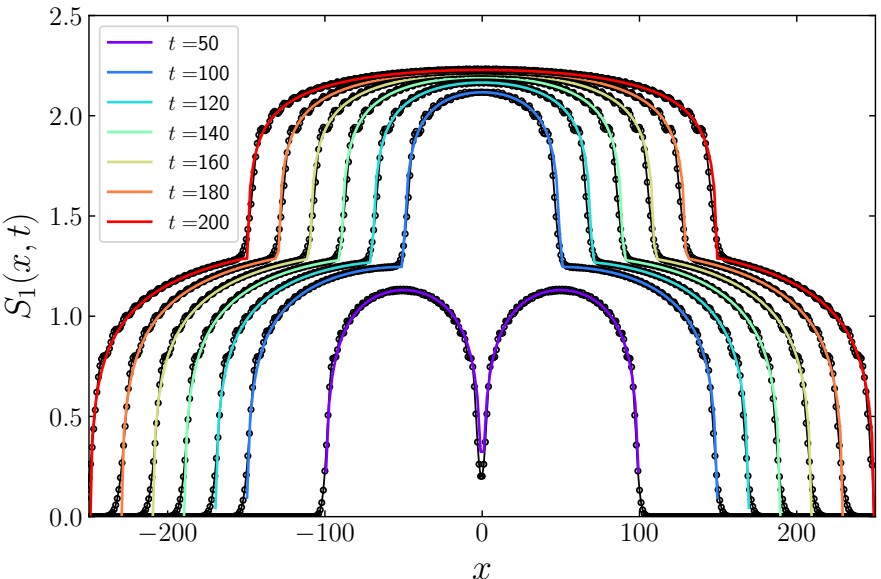

Figure 6: Evolution of the entanglement profiles in the double domain wall melting of a non-interacting spin-1/2 chain. The symbols are obtained with exact lattice numerical calculations while the full-line is our result in Eq. (45). The numerical data are obtained for a chain of size $L = 600$ and setting $\ell = 100$.

where $\Upsilon$ is a non-universal constant [76], related to $C$ as

$$\Upsilon = -\frac{1}{6}\log(C/2) \simeq 0.4785. \tag{46}$$

In the regime $|t - \ell| < |x| \leq t + \ell$ where the two lightcones do not intersect, we observe that Eq. (45) is in agreement with the exact prediction for the entanglement spreading of a DW state, obtained in Ref. [7]. On the contrary, for $|x| \leq t - \ell$, we find a non-trivial behavior of the entanglement that cannot be reduced to just a sum of the two individual contributions coming from each lightcone.

We tested our formula (45) for the entanglement entropy against exact lattice numerical calculations finding an excellent agreement, see Fig. 6. The numerical data are obtained from the two-point correlation matrix in Eq. (24) restricted to an interval $A$ of length $|A|$

$$G_A(t) = \left[ G_{x,x'}(t) \right]_{x,x' \in A}, \tag{47}$$

as [78–83]

$$S_A(t) = \sum_{j=1}^{|A|} \left[ \xi_j(t) \log \xi_j(t) - (1 - \xi_j(t)) \log(1 - \xi_j(t)) \right], \tag{48}$$

where $\xi_j(t)$, $j = 1, \ldots, |A|$ are the eigenvalues of $G_A(t)$, see e.g. Ref. [6,20] for more details.

It is interesting to consider the dynamics of the half-system entanglement by setting $x = 0$ in (45). In this case, the variables in Eq. (40) simplifies to $\varphi = -\phi = \arcsin(\ell/t)$ and the half-system entanglement entropy reads

$$S_1(0, t \geq \ell) = \frac{1}{6}\log\left(\ell^2 \left|1 - \ell^2/t^2\right|^3\right) + 2\Upsilon \overset{t \gg \ell}{\approx} \frac{1}{3}\log\ell + 2\Upsilon. \tag{49}$$

Moreover, by expanding Eq. (45) for $t \gg |x|, \ell$ at leading order, it is easy to show that

$$S_1(x, t) \overset{t \gg |x|, \ell}{\sim} \frac{1}{3}\log\ell + 2\Upsilon. \tag{50}$$

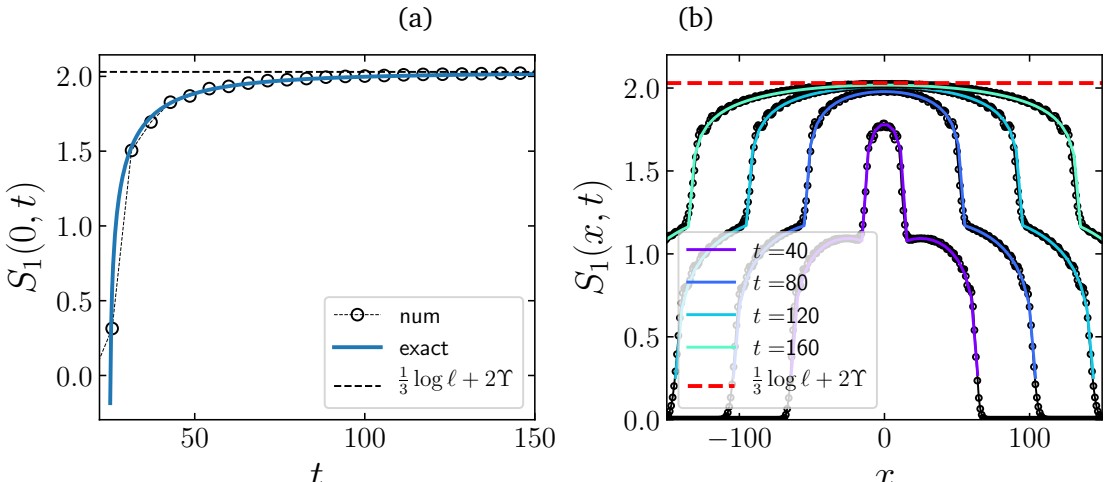

Figure 7: (a) Half-system entanglement entropy for a non-interacting spin-1/2 chain initially prepared in a double domain wall state (4) as function of time. The exact prediction in Eq. (49) (full-line) is compared with the numerical data (symbols) obtained for a lattice of size $L = 350$ with $\ell = 25$. At times $t \gg \ell$, the half-system entanglement saturates to the constant value $\frac{1}{3}\log\ell + 2\Upsilon$ (dashed horizontal line). (b) Asymptotic saturation of the entanglement profiles to the value $\frac{1}{3}\log\ell + 2\Upsilon$ for spatial positions $|x| \ll t$, see Eq. (50).

The saturation of entanglement in Eq. (49)-(50) can understood with a simple CFT argument that we now discuss. At large but fixed time $t \gg |x|, \ell$, the bulk of the system is characterized by a homogeneous magnetization profile (see Eq. (21))

$$m(x, t) \overset{t \gg |x|, \ell}{\sim} -\frac{1}{2} + \frac{2\ell}{\pi t} \tag{51}$$

and by a correlated region of size (see Fig. 3(a))

$$\mathcal{L}(t) = 2(t - \ell) \overset{t \gg \ell}{\sim} 2t. \tag{52}$$

This asymptotic setup can be equivalently interpreted as a homogeneous finite-size system of length $\mathcal{L}(t)$ containing two species of fermionic particles (one from each junction) with density $\varrho = (m + 1/2)/2 \sim \ell/(\pi t)$, whose half-system entanglement is known [73,84] and reproduces the result in Eq. (50)

$$S_1(0, t) \sim \frac{c}{6}\log\left(\frac{\mathcal{L}(t)}{2\epsilon(t)}\right) = \frac{1}{3}\log\ell + 2\Upsilon, \tag{53}$$

after setting $c = 2$ and the short distance cutoff to $\epsilon(t) = C/\sin(\pi\varrho) \sim Ct/\ell$ (see Eq. (42) and Ref. [76]). In Fig. 7, we compare the results in Eq. (49)-(50) with exact lattice calculations of the half-system entanglement. As one can see, our exact formula is in perfect agreement with the numerical data. Notice that the result in Eq. (50) agrees with that of Ref. [85], obtained with exact lattice calculations. This gives a non-trivial check about the validity of our hydrodynamic approach.

## 4 Analytic solution of the interacting case

We now turn to the discussion of a gapless XXZ model (1) for general $|\Delta| < 1$. As we argue below, the presence of non-zero interactions requires some adjustments in our hydrodynamic

description but it does not qualitatively affect the physical dynamics obtained from the solution of the non-interacting limit in Sec. 3.

## 4.1 Bethe Ansatz solution

As anticipated in Sec. 2, the Hamiltonian (1) is diagonalized by means of the Bethe Ansatz [55,56]. In particular, the eigenstates of the model can be written down exactly and are labeled by a set of complex numbers $\{\lambda_j\}$ (also called *rapidities*) which generalize the concept of particle momenta of a free Fermi gas to the interacting case. The value of $\{\lambda_j\}$ is not arbitrary but it is found as solution of a set of non-linear algebraic equations called Bethe Ansatz equations. In the Hilbert space sector with $M$ spins up and assuming periodic boundary conditions of the chain, the Bethe Ansatz equations take the form [55]

$$\left(\frac{\sinh(\lambda_j + \mathbf{i}\gamma/2)}{\sinh(\lambda_j - \mathbf{i}\gamma/2)}\right)^L = \prod_{1 \le i < j \le M} \frac{\sinh(\lambda_j - \lambda_i + \mathbf{i}\gamma)}{\sinh(\lambda_j - \lambda_i - \mathbf{i}\gamma)}, \qquad \gamma = \arccos\Delta \in (0,\pi). \tag{54}$$

Eq. (54) implements non-trivial quantization conditions for the interacting model (1) on a ring of length $L$. In the thermodynamic limit $L \to \infty$ and according to the string hypothesis [55], the solutions of Eq. (54) are arranged into strings i.e., in regular patterns in the complex plane composed by $l_j$ rapidities having the same real part $\lambda_j^\beta$ and equidistant imaginary parts

$$\{\lambda_j\}_{j=1}^M \;\to\; \left\{\lambda_j^\beta + \mathbf{i}\frac{\gamma}{2}\left(l_j + 1 - 2r\right) + \mathbf{i}\frac{\pi(1-u_j)}{4}\right\} \quad \begin{matrix} j = 1,\dots,\delta; \\ r = 1,\dots,l_j; \\ u_j = \pm 1. \end{matrix} \tag{55}$$

Here, $\delta$ is the total number of strings (given in Eq. (3)), $l_j$ is the length and $u_j$ is the parity of a given string while $\beta$ is a label for the strings of a given species, see Appendix A for more details and e.g. Ref. [55,56] for a more systematic treatment. Deviations from Eq. (55) are expected to vanish exponentially with the system size when $L \to \infty$. The different strings are interpreted as $\delta$ different species of quasiparticles, each described by a real rapidity $\lambda_j^\beta$, corresponding to the string center. Since for $L \to \infty$ the spectrum becomes densely populated, it is useful to introduce a macroscopic spectral distribution of quasiparticles

$$\rho_j(\lambda_j^\beta) = \lim_{L\to\infty} \frac{1}{(L|\lambda_j^{\beta+1} - \lambda_j^\beta|)} \tag{56}$$

that satisfy the following integral equation [55]

$$s_j \rho_j(\lambda) = a_j(\lambda) - \sum_{k=1}^\delta \int_{-\infty}^\infty d\mu \; T_{j,k}(\lambda - \mu)\, \rho_k(\mu), \tag{57}$$

where

$$a_j(\lambda) = \frac{u_j}{\pi} \frac{\sin(\gamma l_j)}{\cosh(2\lambda) - u_j \cos(\gamma l_j)} \equiv a_{l_j}^{(u_j)}(\lambda), \tag{58a}$$

$$T_{j,k}(\lambda) = (1 - \delta_{l_j,l_k})a_{|l_j-l_k|}^{(u_j u_k)}(\lambda) + 2a_{|l_j-l_k|+2}^{(u_j u_k)}(\lambda) + \cdots + 2a_{l_j+l_k-2}^{(u_j u_k)}(\lambda) + a_{l_j+l_k}^{(u_j u_k)}(\lambda) \tag{58b}$$

and we recall that $s_j$ denotes the sign of the string (see Appendix A). The solution of Eq. (57) allows us to characterize the ground state of a translationally-invariant Hamiltonian (1) in the thermodynamic limit. For more generic states, the l.h.s. of Eq. (57) is modified as $\rho_j \to \rho_j + \rho_j^h$,

with $\rho_j^h$ a distribution function for the unoccupied quasimomenta. The relation between $\rho_j$ and $\rho_j^h$ is not determined by Bethe Ansatz alone but it requires additional thermodynamic arguments [55]. For later convenience, we introduce also the occupation function

$$\mathrm{n}_j(\lambda) = \frac{\rho_j(\lambda)}{\rho_j(\lambda) + \rho_j^h(\lambda)}, \tag{59}$$

which will play the same role of the Wigner function of Sec. 3.1 for the interacting model (1) in its hydrodynamic description, see Sec. 4.3.

## 4.2 Large-scale description of the initial state

Let us now consider a large-scale asymptotic description of the model (1) in terms of periodic boxes $\Delta x$, each containing a large number of lattice sites. In this way, we can determine the local macrostate $\mathrm{n}_j(x, \lambda)$ by solving the TBA equations of Sec. 4.1 within each coarse-grained cell $x$. At this point, we are in need of a large-scale description of the macrostate corresponding to the intial state in Eq. (4). To achieve this goal, we first investigate the TBA solution of a generic spin-state

$$\hat{\varrho}(h) = \exp\left(2h\hat{S}_z\right)/\mathcal{Z}; \quad \mathcal{Z} = \mathrm{tr}\left(e^{2h\hat{S}_z}\right), \tag{60}$$

where $h$ is an external magnetic field and $\hat{S}_z = \sum_{x=-L/2}^{L/2} \hat{\sigma}_x^z$ is the total spin in the z-direction. In this simple instance, the occupation functions are known exactly [40, 55]. In particular, the first $\delta - 2$ strings are

$$\mathrm{n}_j^{(h)}(\lambda) = \left[\frac{\sin(y_i h)}{\sinh((l_j + y_i)h)}\right]^2, \qquad m_i \le j < m_{i+1}; \ j \le \delta - 2, \tag{61}$$

where $\{y_{-1}, y_0, \ldots, y_q\}$ and $\{m_0, m_1, \ldots, m_q\}$ are two set of numbers related to $\{v_1, \ldots, v_q\}$, as reported in Appendix A. The second-to-last and the last strings have instead a different behavior

$$\mathrm{n}_{\delta-1}^{(h)}(\lambda) = \frac{1}{1 + \kappa e^{hP}}, \quad \mathrm{n}_\delta^{(h)}(\lambda) = \frac{\kappa}{1 + \kappa e^{hP}}; \qquad \kappa = \frac{\sin(l_{\delta-1}h)}{\sin(l_\delta h)}. \tag{62}$$

In the limit of strong magnetic field $|h| \to \infty$, the spin-state (60) becomes fully-polarized and it reduces to the projector

$$\lim_{|h| \to \infty} \hat{\varrho}(h) = \begin{cases} \hat{\varrho}^{(\Uparrow)} \equiv |\Uparrow\rangle \langle \Uparrow|, & \text{if } h > 0; \\ \hat{\varrho}^{(\Downarrow)} \equiv |\Downarrow\rangle \langle \Downarrow|, & \text{otherwise}, \end{cases} \tag{63}$$

where $|\Uparrow\rangle \equiv \bigotimes_x |\uparrow_x\rangle$ and similarly for $|\Downarrow\rangle$. As a consequence, the occupation functions in Eq. (61)-(62) simplify to

$$\lim_{h \to \infty} \mathrm{n}_j^{(h)} = \begin{cases} \delta_{j,\delta} + \delta_{j,\delta-1}, & \text{if } h > 0; \\ 0, & \text{otherwise}, \end{cases} \tag{64}$$

signaling that only the largest two strings are responsible for the thermodynamics of the polarized state (63) [40, 41]. This observation is at the basis of our analytical solution of the double domain wall melting problem, as discussed in the following section.

From the solution (64) of a polarized spin state, we can easily derive the occupation functions $\mathrm{n}_j(x, \lambda)$ corresponding to the double DW state (4). In particular, we associate to each coarse-grained cell $x$ the polarized state

$$\hat{\varrho}(x) = \begin{cases} \hat{\varrho}^{(\Uparrow)}, & \text{if } |x| \le \ell; \\ \hat{\varrho}^{(\Downarrow)}, & \text{otherwise}, \end{cases} \tag{65}$$

so that the initial occupation functions for our setup read

$$\mathrm{n}_j(x,\lambda) = \Theta(\ell - |x|)(\delta_{j,\delta} + \delta_{j,\delta-1}), \tag{66}$$

with $\Theta$ the Heaviside step-function.

## 4.3 Generalized Hydrodynamics

The hydrodynamic evolution of the initial state (66) is established by means of Generalized Hydrodynamics (GHD),which provides the following set of continuity equations for the occupation functions [35, 42]

$$\left(\partial_t + v_j^{\mathrm{eff}}\partial_x\right)\mathrm{n}_j(x,t,\lambda) = 0. \tag{67}$$

This set of equations has the same structure of Eq. (12), obtained in Sec. 3.1 for the non-interacting model, but it is characterized by an effective velocity $v_j^{\mathrm{eff}} = v_j^{\mathrm{eff}}(x,t,\lambda)$

$$v_j^{\mathrm{eff}} = \frac{\partial_\lambda e_j^{\mathrm{dr}}}{\partial_\lambda p_j^{\mathrm{dr}}}, \tag{68}$$

which depends self-consistently on the state $\mathrm{n}_j(x,t,\lambda)$ through the dressing operation [35, 40]

$$q_j^{\mathrm{dr}\prime}(x,t,\lambda) = q_j'(\lambda) - \sum_{k=1}^{\delta}\int_{-\infty}^{\infty}\mathrm{d}\mu\; T_{j,k}(\lambda-\mu)\, s_k\, \mathrm{n}_k(x,t,\mu)\, q_k^{\mathrm{dr}\prime}(x,t,\mu), \tag{69}$$

where $q_j$ is a generic function[1]. For the specific case of Eq. (68), the energy $e(\lambda)$ and the derivative of the momentum eigenvalue $p'(\lambda)$ are

$$e(\lambda) = -\pi\sin(\gamma)a_j(\lambda), \qquad p'(\lambda) = 2\pi a_j(\lambda) \tag{70}$$

and their dressed value is obtained as solution of Eq. (69).

The GHD equations (67) are exact only in the limit of large space and time scales. Nevertheless, they proved to give a very accurate description of the dynamics even for relatively small times and distances, as confirmed by several numerical tests e.g. [35, 37, 40, 42, 89–93] and even some experimental studies [52, 94, 95]. Therefore, we will make use of the GHD equations as a basis for our analysis, providing further numerical tests based on time-dependent Density Matrix Renormalization Group (tDMRG) simulations of our results.

A formal solution of GHD equations (67) is obtained with method of characteristics, yielding

$$\mathrm{n}_j(x,t,\lambda) = \mathrm{n}_j(\tilde{x}(t),0,\lambda), \tag{71}$$

where

$$\tilde{x}(t) = x - \int_0^t \mathrm{d}s\; v_j^{\mathrm{eff}}(s,\tilde{x}(s),\lambda). \tag{72}$$

Plugging the initial state (66) in Eq. (71), it is easy to see that only on the largest two strings are relevant for the dynamics since

$$\mathrm{n}_j(x,t,\lambda) = \Theta(\ell - |\tilde{x}(t)|)(\delta_{j,\delta} + \delta_{j,\delta-1}). \tag{73}$$

Moreover, the structure of the interaction kernels in Eq. (58) reveals the following symmetry properties of the last and the second-to-last strings [40]

$$a_\delta(\lambda) = -a_{\delta-1}(\lambda), \qquad T_{\delta,k}(\lambda) = -T_{\delta-1,k}(\lambda), \tag{74}$$

---

[1] See also Ref. [86–88] for a rigorous proof of Eq. (67), (68).

and the signs $s_\delta = -s_{\delta-1}$. This implies that the dressing operation (69) of the largest two strings gives

$$p_\delta^{dr\prime}(\lambda) = \; 2\pi a_\delta(\lambda) - \int_{-\infty}^{\infty} d\mu \; T_{\delta,\delta}(\lambda - \mu) s_\delta \Big( n_\delta(x,t,\mu) \, p_\delta^{dr\prime}(x,t,\mu) \tag{75}$$

$$+ n_{\delta-1}(x,t,\mu) \, p_{\delta-1}^{dr\prime}(x,t,\mu) \Big) = -p_{\delta-1}^{dr\prime}(\lambda) \tag{76}$$

and, with the same calculation, one can show that $e_\delta^{dr\prime} = -e_{\delta-1}^{dr\prime}$. It follows that the largest two strings share the same effective velocity (68) $v_\delta^{eff} = v_{\delta-1}^{eff}$, which in turns implies from Eq. (73) that the occupation functions

$$n_\delta(x,t,\lambda) = n_{\delta-1}(x,t,\lambda). \tag{77}$$

From this observation, using Eq. (77) in Eq. (75) (and similarly for the dressed energy), we conclude that the dressing operation becomes trivial and the effective velocity $v_\delta^{eff}$ reduces to

$$v_\delta^{eff} \equiv v_\delta = \frac{\partial_\lambda e}{\partial_\lambda p} = \zeta_0 \sin(s_\delta \, p_\delta(\lambda)), \tag{78}$$

where we defined

$$\zeta_0 = \sin\gamma / \sin(\pi/P). \tag{79}$$

Therefore, plugging Eq. (78) into Eq. (73), we obtain the following analytical solution of the GHD equations (67)

$$n_j(t,x,\lambda) = \Theta(\ell - |x - v_j(\lambda)t|, \lambda), \qquad j \in \{\delta-1, \delta\}. \tag{80}$$

## 4.4  Magnetization and Spin-current profiles

The analytic solution (80) of GHD allows us to derive exact asymptotic results for the conserved charges and currents of the model. Proceeding similarly to what done in Sec.3.1, we define the (split-) Fermi sea of the interacting model as

$$\Gamma(x,t) = \{\lambda : \; x = \pm\ell + v_\delta(\lambda)t\} . \tag{81}$$

This equation of motion can be easily solved by noticing that the quantity $s_\delta p_\delta(\lambda)$ is a monotonic function in the interval $[-\pi/P, \pi/P]$. Therefore, in analogy with the free case, we define the function $k(\lambda) \equiv s_\delta p_\delta(\lambda)$ and we obtain the roots

$$k_F = \left\{ \arcsin\frac{x \mp \ell}{t\zeta_0} \; ; \pi - \arcsin\frac{x \mp \ell}{t\zeta_0} \right\} \tag{82}$$

that arrange together to give the following Fermi points

$$k_F^{(s)}(x,t) = \begin{cases} \pm \arcsin\frac{||x|-\ell|}{\zeta_0 t} ; \pm(\pi - \arcsin\frac{||x|-\ell|}{\zeta_0 t}), \\ \quad \text{if } |\zeta_0 t - \ell| < |x| \le \zeta_0 t + \ell ; \\ \pm \arcsin\frac{|x|+\ell}{\zeta_0 t} ; \pm(\pi - \arcsin\frac{|x|+\ell}{\zeta_0 t}), \mp \arcsin\frac{||x|-\ell|}{\zeta_0 t} ; \mp(\pi + \arcsin\frac{||x|-\ell|}{\zeta_0 t}); \\ \quad \text{if } |x| \le \zeta_0 t - \ell, \end{cases} \tag{83}$$

with $\text{sign}(k_F^{(s)}(x,t)) = \text{sign}(x(|x| - \ell))$. We use then the Fermi points (83) to construct the (split-)Fermi sea $\Gamma(x,t)$ as in Eq. (19). Notice that the structure of $\Gamma(x,t)$ is very similar to that found in the non-interacting case (cf. Eq. (17)) although the presence of the interactions is responsible for a rescaling of the lightcones with $\zeta_0$.

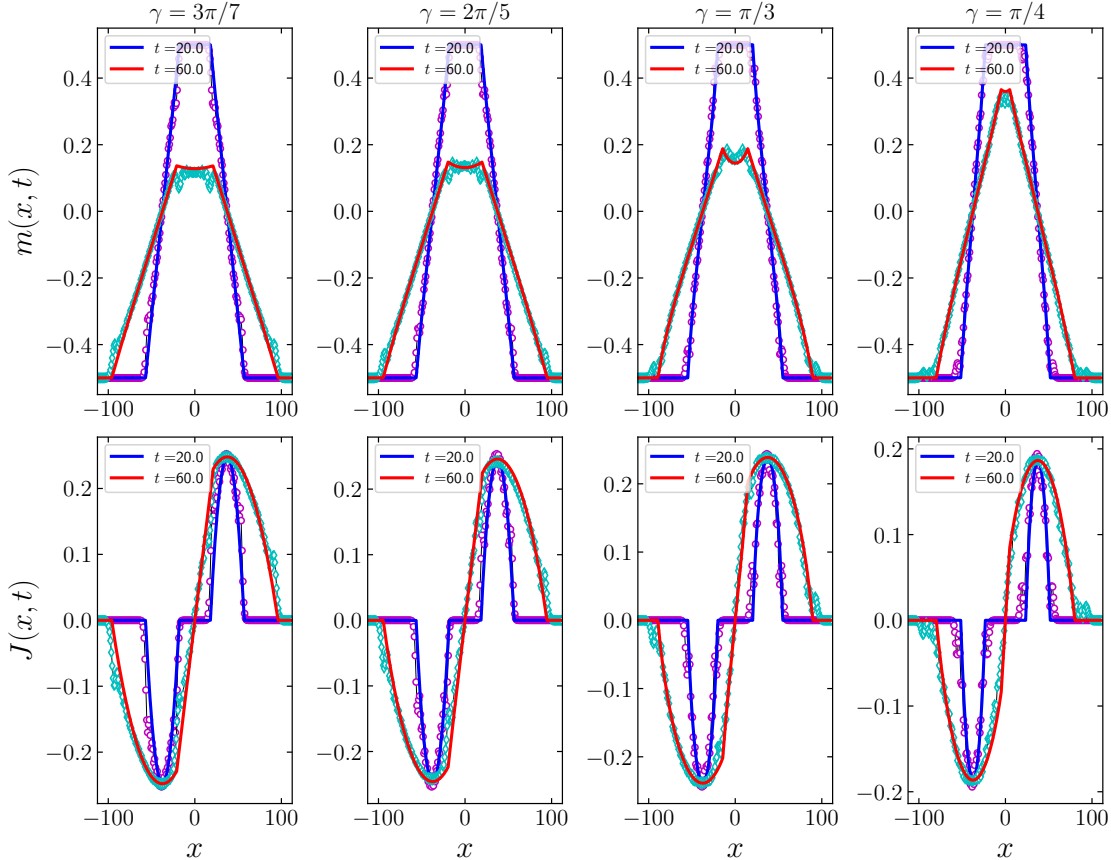

Figure 8: (Top) Magnetization and (bottom) spin current profiles during the double domain wall melting dynamics for several values of the interaction coupling $\gamma$ (corresponding to $\Delta = 0.223, 0.309, 0.5, 0.707$, from the left to the right panels). The different curves show the behavior of the analytic solutions (21)-(22) at different times while the symbols are obtained with tDMRG numerical simulations obtained for a lattice of size $L = 225$ and $\ell = 35$. The matching of the profiles with the numerics is seen to be very good although finite-size effects are still visible.

At this point, we are ready to compute the non-equilibrium dynamics of the charges profiles. In particular, the profile of the magnetization is obtained as the weighted sum over the single quasiparticle contributions

$$m(t,x) = -\frac{1}{2} + \sum_{j=1}^{\delta} \int_{-\infty}^{\infty} \frac{d\lambda}{2\pi} s_j \, p_j^{\mathrm{dr}\prime}(x,t,\lambda) \, \mathrm{n}_j(x,t,\lambda) \tag{84}$$

that, in the case of a double DW melting dynamics, reduces to

$$m(x,t) = -\frac{1}{2} + \int_{\Gamma(x,t)} d\lambda \, \frac{k'(\lambda)}{2\pi}$$

$$= \begin{cases} -\frac{1}{2} + \frac{P}{2\pi} \left( \arcsin \frac{|x|+\ell}{\zeta_0 t} - \arcsin \frac{|x|-\ell}{\zeta_0 t} \right), \\ \qquad \text{if } |x| \leq \zeta_0 t - \ell\,; \\ -\frac{P}{2\pi} \arcsin \frac{|x|-\ell}{\zeta_0 t}\,, \qquad \text{if } |\zeta_0 t - \ell| < |x| \leq \zeta_0 t + \ell\,. \end{cases} \tag{85}$$

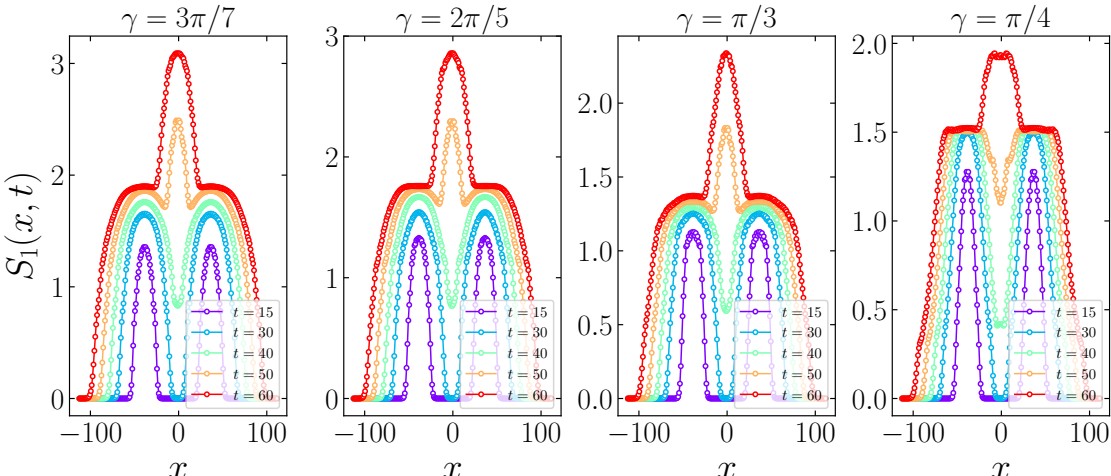

Figure 9: Snapshots of the entanglement profiles during the double DW melting dynamics for different values of the interaction coupling $\gamma$. The data are obtained with tDMRG for a system of size $L = 225$ and $\ell = 35$, and correspond to the charges profiles shown in Fig. 8

Similarly, for the spin current one obtains the generic expression

$$J(t,x) = \sum_{j=1}^{\delta} \int_{-\infty}^{\infty} \frac{\mathrm{d}\lambda}{2\pi} s_j \, p_j^{\mathrm{dr}'}(x,t,\lambda) \, v_j^{\mathrm{eff}}(x,t,\lambda) \, \mathrm{n}_j(x,t,\lambda) \tag{86}$$

that simplifies, thanks to the solution in Eq. (83), to

$$J(x,t) = \int_{\Gamma(x,t)} \frac{\mathrm{d}\lambda}{2\pi} \, k'(\lambda) \, \zeta_0 \sin(k(\lambda))$$

$$= \begin{cases} \frac{\mathrm{sgn}(x)\zeta_0}{2\pi/P} \left( \sqrt{1 - \frac{(|x|-x_0)^2}{\zeta_0^2 t^2}} - \sqrt{1 - \frac{(|x|+x_0)^2}{\zeta_0^2 t^2}} \right), & \text{if } |x| \le \zeta_0 t - \ell \, ; \\ \frac{\mathrm{sgn}(x)\zeta_0}{2\pi/P} \left( \sqrt{1 - \frac{(|x|-x_0)^2}{\zeta_0^2 t^2}} - \cos(\pi/P) \right), & \text{if } |\zeta_0 t - \ell| < |x| \le \zeta_0 t + \ell \, . \end{cases} \tag{87}$$

Notice that Eqs. (85)-(87) correctly reproduce the non-interacting results of Eqs. (21)-(22) when $Q = 1$, $P = 2$ (i.e., for $\Delta = 0$). As expected, in the regime $|\zeta_0 t - \ell| < |x| \le \zeta_0 t + \ell$, the exact asymptotic formulae (85)-(87) are the same as those of a single DW, derived in Ref. [40]. On the other hand, the analytic solution of the GHD in the regime $|x| \le \zeta_0 t - \ell$ is non-trivial.

We tested our results (85)-(87) with tDMRG numerical simulations performed using the open-source library iTensor [96], as shown in Fig. 8 for several values of the interacting coupling $\Delta$. The physical parameters in the simulations had to be chosen to achieve the best compromise between having small finite size effects (so large $L$ and $\ell$) and little entanglement (so not too long times). The resulting values are those reported in Fig. 8. Indeed, although finite-size effects are still visible, the agreement of the curves with the numerics is very good and it undoubtedly confirms our exact results.

### 4.5 Numerical analysis of the Entanglement entropy

Finally, we discuss the dynamics of the entanglement for the interacting model (1). We notice that a re-quantization of the GHD evolution of Sec. 4.3 in terms of a Luttinger liquid is possible also in the interacting case, as outlined in Ref. [43]. Moreover, in the specific case of a single DW,

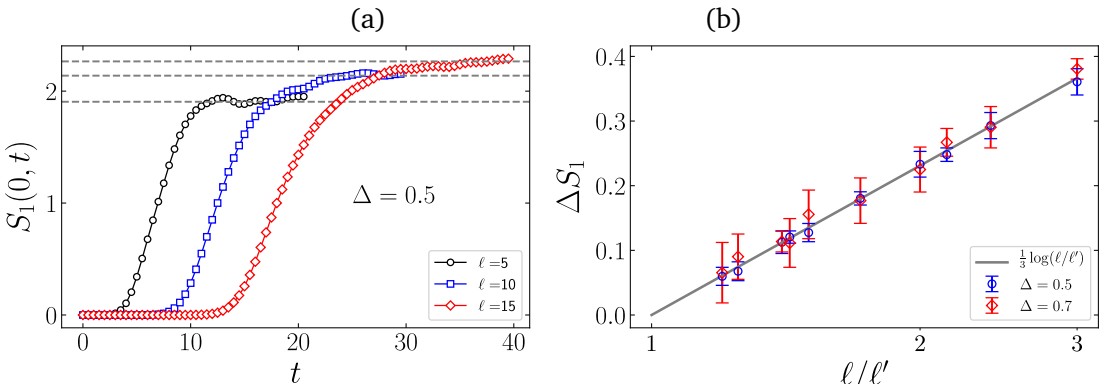

Figure 10: (a) Numerical analysis of the half-system entanglement at large times $t \gg \ell$ for a lattice of size $L = 120$ with $\Delta = 0.5$ and different domain sizes $\ell$. We observe a saturation of the entanglement to a constant value that we extracted with a best fit of our data (dashed-lines). (b) Comparison of the numerical results with the Ansatz in Eq. (88) using the proxy (89) for $\Delta = 0.5, 0.7$. Although finite-size effects affect our data, we see a good agreement, which we expect to improve with larger values of $L$, $\ell$ and $t$. In the plot, error bars are extracted from the covariance matrix of the fitting plateaus.

it has been shown [41] that the low-energy field theory is characterized by a constant Luttinger parameter, which displays a fractal dependence on $\Delta$. In this simple instance, conformal invariance is not broken and an asymptotic prediction for the asymptotic growth of the half-system entanglement can be derived with CFT scaling arguments [41]. Though, in our double DW setting, this is not the case and we find the exact computation of the entanglement highly non-trivial.

Nevertheless, from the numerical study in Fig. 9, we observe that the behavior of the entanglement profiles is not qualitatively influenced by the presence of interactions. In particular, in the regime $|\zeta_0 t - \ell| < |x| \leq \zeta_0 t + \ell$, we observe that the entanglement spreading is given by two independent contributions that are developed around each junction. Conversely, when $|x| \leq t\zeta_0 - \ell$, we find an interplay of the quasiparticles emitted from the two walls, which results into a fast growth of the half-system entanglement entropy, similarly to what observed in Sec. 3.2 for a free model. Motivated by this strict analogy with the free case, we expect an asymptotic saturation of entanglement and we conjecture the following form

$$S_1(x, t) \sim \frac{1}{3} \log \ell + c(\gamma), \tag{88}$$

where $c(\gamma)$ is a non-universal constant depending on the parameter $\gamma$ that we are unable to determine. However, by considering the entanglement evolution of two initial domains of upwards spins with different sizes (respectively $\ell$ and $\ell'$), we can test our conjecture (88) considering the difference

$$\Delta S_1 = S_1^{(\ell)}(0, t) - S_1^{(\ell')}(0, t) \sim \frac{1}{3} \log \frac{\ell}{\ell'}. \tag{89}$$

We show the results of our simulations in Fig. 10. In particular in Fig. 10(a), we observe an asymptotic saturation of the half-system entanglement for different values of $\ell$. The panel 10(b) is instead a numerical test of our Ansatz (88), where we construct our proxy in Eq. (89) with the plateaus obtained as in Fig. 10(a). We find a good agreement for different values of $\Delta$ already for modest values of $L$ and $\ell$.

## 5 Summary and conclusions

We investigated the non-equilibrium dynamics of a spin-1/2 XXZ model (1) at zero temperature, initially prepared in a state with two domain walls (4). This quench setup is a generalization of the widely studied bipartitioning protocols (see e.g. [4–8, 35, 37, 40–42]), where now a central domain of upwards spins $x \in [-\ell, \ell]$ is joined to two semi-infinite down-oriented ferromagnets, respectively on its left $[-L/2, -\ell - 1]$ and right $[\ell + 1, L/2]$ sides. In particular, we focused on the rational case, where $\Delta = \cos \gamma$ and $\gamma = \pi Q/P$, with $Q, P$ two co-prime integers. In this case, as first noticed in Ref. [40], the structure of the Bethe Ansatz solution allows us to analytically solve the GHD equations (67) and, as a consequence, to determine the exact asymptotic evolution of the charges and currents profiles during the melting dynamics (see Eq. (85) and (87)). The non-trivial result of this paper is that such exact solvability of the model extends also to a late-time regime characterized by the presence of correlations between the two domain walls, and therefore, by the emergence of split Fermi-seas. To our best knowledge, this is the first case where a fully-analytical GHD solution is found with multiple Fermi points. In Sec. 3, we also treated the case of a non-interacting spin chain (i.e. a free Fermi gas initially confined in a segment of size $2\ell$) in detail. For the free gas, after the derivation of the charges profile with standard hydrodynamic techniques (see Sec. 3.1), we applied recent developments in quantum fluctuating hydrodynamics [6, 54] to give an exact description of the entanglement entropy (see Eq. (45)). Moreover, we found an asymptotic saturation of the half-system entanglement entropy as $S_1(0, t) \sim \frac{1}{3} \log \ell$. Motivated by the strict analogy of the melting dynamics in the free and interacting case, we expect a similar behavior for the interacting chain (88), as confirmed by several numerical simulations (see Fig. 10).

   We mention that the exact solution of the GHD equations reported in this work can be, in principle, extended to a whole class of spin chain polarized states such as multiple domain wall configurations and spin-helix states [97]. Nonetheless, it could be interesting to investigate the entanglement evolution of a double domain wall state with quantum GHD [41,43] and, in this way, to prove our conjecture (88) about the half-system entanglement growth.

## Acknowledgments

The authors are very thankful to Viktor Eisler for pointing out a mistake in the definition of the additive constant $\Upsilon$ in Eq. (46) appearing in the first version of this work. SS acknowledges Alexandre Krajenbrink for discussions on closely related projects.

**Funding information**   PC and SS acknowledge support from ERC under Consolidator grant number 771536 (NEMO). JD acknowledges support from *CNRS International Emerging Actions* under the grant QuDOD.

## A   Some details on the strings solutions.

In this appendix, we report some useful formulae for the determination of the length $l_j$, parity $u_j$ and sign $s_j$ of a given string solution $j \in \{1, \ldots, \delta\}$, following the discussion of Ref. [55]. From the set of numbers $\{v_1, \ldots, v_q\}$, which are obtained from the continued fraction representation of the interaction coupling $\gamma$ (see Eq. (2)), we construct the two set of numbers $\{y_{-1}, y_0, \ldots, y_q\}$, $\{m_0, m_1, \ldots, m_q\}$ as

$$y_{-1} \equiv 0, \quad y_0 \equiv 1, \quad y_1 = v_1, \quad y_{i \geq 2} = v_i y_{i-1} + y_{i-2}; \tag{90a}$$

$$m_0 \equiv 0; \qquad m_{i \geq 1} = \sum_{p=1}^{i} \nu_p \quad (m_q \equiv \delta). \tag{90b}$$

In terms of these series of numbers we have the following relations for the length $l_j$

$$l_j = y_{i-1} + (j - m_i) y_i \quad \text{for} \quad m_i < j < m_{i+1}, \quad \text{and} \quad l_\delta \equiv y_q; \tag{91}$$

the parity $u_j$

$$u_j = (-1)^{(l_j-1)Q/P} \quad \text{for} \quad m_i < j < m_{i+1}, \quad \text{and} \quad u_\delta \equiv (-1)^q; \tag{92}$$

and the sign $s_j$

$$s_j = \text{sign}(\omega_j), \quad \text{for} \quad m_i \leq j < m_{i+1}, \tag{93}$$

where $\omega_j \equiv (-1)^i (p_i - (j - m_i) p_{i+1})$ and the series $\{p_0, \dots, p_q\}$ is defined by

$$p_0 \equiv \pi/\gamma, \quad p_1 \equiv 1, \qquad p_{i \geq 2} = p_{i-2} - \nu_{i-1} p_{i-1}. \tag{94}$$

As a concrete example, let us consider the case with $Q \equiv 1$ (hence, $\gamma = \pi/P$), known also as *root of unity point*. In this instance, one finds $\delta \equiv P$ strings (see Eq. (3)) with the following properties

$$l_j = j, \quad u_j = 1, \quad \omega_j = P - l_j \quad j \in \{1, \dots, P-1\}; \tag{95}$$

$$l_P = 1, \quad u_P = -1, \quad \omega_P = -1. \tag{96}$$

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
