# Peer review of "Exact hydrodynamic solution of a double domain wall melting in the spin-1/2 XXZ model"

_SciPost Physics, doi:SciPost Phys. 12, 207 (2022)_

## Round 1 · Referee Report · Anonymous (Referee 1) · 2021-11-2

Report
I have a few comments:
p2: “a full analytical understanding of the DW dynamics”: this is a bit of an overstatement. The exact hydrodynamic solutions pertain to a particular scaling limit, and the exact microscopic dynamics for a domain wall with Delta != 0 remains unsolved. This lack of analytical understanding is particularly pronounced at Delta = 1. The front dynamics subleading to the Euler hydrodynamics (studied numerically e.g. in JM Stephan, SciPostPhys.6.5.057) is also not understood analytically for Delta != 0.
p2: “Quite interestingly, the physics of the melting process is not modified by the presence of interactions.” Again, this is only true at the ballistic scale and for |Delta| < 1. As discussed in SciPostPhys.6.5.057, interactions lead to qualitatively different front scaling for |Delta| > 0 compared to Delta=0. Interactions also (obviously) dictate the dynamics for |Delta|>= 1. For a finite density of domain walls at |Delta| > 0, interactions should furthermore eventually thermalize the system to a GGE (more on this below).
p6, eq. 12: the justification for this equation provided in the text is a bit ad hoc, given that the “Wigner function approach” is referred to specifically in the abstract – it might be worth noting explicitly in the text that this equation was derived microscopically for the XX chain in Refs. 59-60.
Sec 3.2 and 4.5: The scaling of leading-order corrections to GHD in integrable systems is reasonably well-understood by now. What is less clear to me is the regime of validity of the procedure used by the authors, when they state that the “asymptotic behavior of the entanglement is effectively described by a quantum hydrodynamic theory, obtained after the re-quantization of the Fermi contour of Sec. 3.1 in terms of a Luttinger-liquid”. While the agreement of the authors' predictions with numerics is impressive, the formalism of Ref. 43 makes several more approximations than ballistic GHD. Could the authors discuss in the text what these approximations underlying QGHD are and why they are so accurate for the quench under consideration? e.g. how do the leading corrections to QGHD scale in space and time?
In particular, while a lack of entropy growth is plausible for Delta = 0, for |Delta| > 0 any finite density of domain walls eventually generates entropy via quasiparticle diffusion, which invalidates the assumptions of Ref. 43, even though the initial state has zero entropy (in a particular coarse-graining). So it seems that “diluteness” of the two-wall initial condition is important for the validity of the authors’ proposal.
Finally, I noticed the following typos:
p4: address the reader -> refer the reader?
p8: contour ad -> contour and?
p12: Fisher-Hartwing - > Fisher-Hartwig
We thank the referee for her/his/their careful reading of our manuscript and for her/his/their positive assessment. In the following, we reply point-by-point to the report.
>>p2: “a full analytical understanding of the DW dynamics”: this is a bit of an overstatement. The exact hydrodynamic solutions pertain to a particular scaling limit, and the exact microscopic dynamics for a domain wall with Delta != 0 remains unsolved. This lack of analytical understanding is particularly pronounced at Delta = 1. The front dynamics subleading to the Euler hydrodynamics (studied numerically e.g. in JM Stephan, SciPostPhys.6.5.057) is also not understood analytically for Delta != 0.
We agree with the referee that the sentence in p2 of the introduction is incorrect outside of the regime $|\Delta|<1$ considered in the main text. Therefore, in the revised version of Sec. 1, we explicitly refer to the XXZ model with anisotropy parameter $|\Delta|<1$. We also added a sentence about the other regimes of the XXZ model (that the referee mentioned) where a full analytical understanding is still lacking. These changes are highlighted in the text.
>>p2: “Quite interestingly, the physics of the melting process is not modified by the presence of interactions.” Again, this is only true at the ballistic scale and for |Delta| < 1. As discussed in SciPostPhys.6.5.057, interactions lead to qualitatively different front scaling for |Delta| > 0 compared to Delta=0. Interactions also (obviously) dictate the dynamics for |Delta|>= 1. For a finite density of domain walls at |Delta| > 0, interactions should furthermore eventually thermalize the system to a GGE (more on this below).
Similarly to the previous point, we agree with the referee’s comment. Since in the revised version the statements in Sec. 1 refer only to the case $|\Delta|<1$, we do not see the need of further changes in the text. Higher-order corrections to GHD are neglected, we specified this in the text by writing: “This exact solution substantiates an intuitive hydrodynamic picture at ballistic scales [...]”
>>p6, eq. 12: the justification for this equation provided in the text is a bit ad hoc, given that the “Wigner function approach” is referred to specifically in the abstract – it might be worth noting explicitly in the text that this equation was derived microscopically for the XX chain in Refs. 59-60.
In the manuscript, we proposed a short and intuitive justification of Eq. (12-13). With the term “Wigner function approach” that is appearing in the abstract of our manuscript, we wish to refer to the semi-classical evolution of the occupation function in the phase space and to the GHD solution for conserved charges and currents that follows, widely discussed in Sec. 3.1. In the revised version of the manuscript, we complemented our justification of Eq.(12-13) with an explicit reference to Ref. [60-61] where the reader can find a more rigorous microscopic derivation. Changes are highlighted in the text.
>>Sec 3.2 and 4.5: The scaling of leading-order corrections to GHD in integrable systems is reasonably well-understood by now. What is less clear to me is the regime of validity of the procedure used by the authors, when they state that the “asymptotic behavior of the entanglement is effectively described by a quantum hydrodynamic theory, obtained after the re-quantization of the Fermi contour of Sec. 3.1 in terms of a Luttinger-liquid”. While the agreement of the authors' predictions with numerics is impressive, the formalism of Ref. 43 makes several more approximations than ballistic GHD. Could the authors discuss in the text what these approximations underlying QGHD are and why they are so accurate for the quench under consideration? e.g. how do the leading corrections to QGHD scale in space and time?
We thank the referee for the very valid question. The re-quantization of GHD in terms of a Luttinger liquid theory has been discussed in more details in the original paper of QGHD, PRL 124, 140603 (2020) where it was shown that this choice leads to the correct Heisenberg equation for the quantized sound waves arising around the Fermi contour. Although QGHD can be seen as a reasonable outcome obtained by combining together linear fluctuating hydrodynamics and Haldane’s quantum fluid theory, it is true that the corrections to QGHD are not yet quantified and therefore a numerical check of QGHD predictions is highly desiderable. We are planning to investigate such corrections to QGHD in a future publication since this topic goes far beyond the purposes of the manuscript under review. However, we do expect QGHD to be asymptotically exact at zero temperature and at ballistic scales, i.e., $x, t\to\infty$ at fixed ratio $x/t$.
We would like to add that, in the specific case of the double domain wall melting, in Ref. 85 was found the same asymptotic behavior of entanglement in Eq.(50) from an exact lattice calculation, giving a non-trivial check of the validity of QGHD approach.
Concluding, we also mention that the impressive agreement with numerics is not specific of domain wall quenches but it has been observed in other protocols with hard-core particles, see e.g. J. Phys. A: Math. Theor. 55, 024003 (2022).
>>In particular, while a lack of entropy growth is plausible for Delta = 0, for |Delta| > 0 any finite density of domain walls eventually generates entropy via quasiparticle diffusion, which invalidates the assumptions of Ref. 43, even though the initial state has zero entropy (in a particular coarse-graining). So it seems that “diluteness” of the two-wall initial condition is important for the validity of the authors’ proposal.
Our discussion of the entanglement in the case $0<|\Delta|<1$ is based on time-dependent DMRG simulations. The latter reveal a behavior of entanglement that is qualitatively similar to that found with QGHD in the non-interacting case. In light of this, we expect that a QGHD analysis of the entanglement as in PRB 102, 180409 (2020) will give a reliable description of the numerical data, signaling that the diffusive correction have a subleading contribution.
>>Finally, I noticed the following typos:
p4: address the reader -> refer the reader?
p8: contour ad -> contour and?
p12: Fisher-Hartwing - > Fisher-Hartwig
We thank the referee for bringing to our attention these typos that we readily fixed in the text.

Author: Stefano Scopa on 2022-01-10 [id 2084]
(in reply to Report 2 on 2021-12-04)We thank the referee for the positive assessment on our manuscript. In the following, we reply to her/his/their comment:
We are not sure to understand the referee’s comment. In particular, in the double domain wall considered in this work, we remain at any time in a zero-entropy state even when a split Fermi sea is generated i.e., the occupation function $n_j$ takes only the values {0,1}, see Eq.(80). More precisely, in the manuscript we consider an initial state with zero-entropy (in our case, a product state) and we evolve it with a Euler hydrodynamic equation (Eq.~(12)/(67)) . This choice for the initial state and for the dynamics ensures that the system is found in a zero-entropy state at all times, see PRL 119, 195301 (2017). This is not a peculiarity of the domain wall setting but it applies to any – inhomogeneous but locally integrable – system evolved with GHD, even to those systems that develop a split Fermi sea during their evolution. The diffusive corrections in the form of Ref. PRL 121, 160603 (2018) vanish when evaluated in zero-entropy states.
We also notice that a plot of the occupation functions during the melting dynamics would be dependent on the hydrodynamic equation that we choose (in our case, the zero-entropy GHD in the Euler regime) and therefore it is inadequate to observe high-order corrections to ballistic hydrodynamics.
Perhaps the referee has higher order corrections in mind, such as the one appearing in the Moyal expansion for the Wigner function dynamics. In the non-interacting case, the first such correction is known and is proportional to the cubic derivative. The latter have been neglected in our treatment although the good agreement of our analytical formulae with the numerical simulations show that these do not significantly affect the behavior of the system in the regimes that we explored.

---

## Round 1 · Referee Report · Anonymous (Referee 2) · 2021-12-4

Strengths
- exact calculations
- analytical solution of a non-trivial GHD evolution
- entanglement entropy is extracted and well compare with numerics
Weaknesses
- more explications needed in the interacting case and lack of discussion on higher order effects
Report

---

## Round 2 · Author Response

Dear Editor(s),

We thank you for considering our manuscript “Exact hydrodynamic solution of a double domain wall melting in the spin-1/2 XXZ model” for publication in SciPost.
We are pleased to see a positive feedback on our results from both the referees. You can find attached a point-by-point response to the referees reports and a list of changes in the revised version of the manuscript. Changes in the text are highlighted to help the review process.
However, we disagree with the fact that our manuscript does not meet the criteria for publication in SciPost Physics and has been downgraded to SciPost Physics Core. On the basis of the very positive referees reports for SciPost Physics, the motivations for this downgrade are obscure to us.
We then resubmit the manuscript to SciPost Physics being confident that this revised version of the manuscript can meet its acceptance criteria.

Sincerely,

The authors.

---

## Round 2 · List of Changes

We list below the changes that we made on the revised version of the manuscript (changes are highlighted in the text.) 1) We modified the layout of our manuscript using the template of SciPost journals. 2) We modified some sentences in Sec. 1 according to the comment of the referee 1. 3) We inserted Ref. [45]. 4) We added a sentence in Sec. 3.1 below Eq. (13) according to the comment of the referee 1. 5) We corrected small typos in the text, including those mentioned by referee 1.

---

## Editorial Decision

published